# Comparison between Post-Operative Analgesic Efficacy of Low-Concentration High-Volume and High-Concentration Low-Volume Combinations of Ropivacaine for Transverse Abdominis Plane Block in Pediatric Open Inguinal Hernia Repair

**DOI:** 10.3390/jcm8081133

**Published:** 2019-07-30

**Authors:** Hyungmook Lee, Jaehee Chung, Minsoo Lee, Sungwon Yang, Haejin Lee

**Affiliations:** Department of Anesthesiology and Pain medicine, Seoul St. Mary’s Hospital, College of Medicine, The Catholic University of Korea, Seoul 06591, Korea

**Keywords:** inguinal hernia, pediatric, ropivacaine, transverse abdominis plane block

## Abstract

Transverse abdominis plane (TAP) block can provide post-operative analgesia in children undergoing open inguinal hernia repair. However, the optimal anesthetic dose, and concentration for TAP block in the pediatric population, is not well defined. This study compared the post-operative analgesic effect of TAP block between low-concentration, with high-volume (LCHV) and high-concentration, with low-volume (HCLV) combinations of local anesthetic. Forty-four patients who underwent open inguinal hernia repair were randomly assigned to two groups. The patients in the LCHV group received 0.67 mL/kg of 0.15% ropivacaine. Whereas, those in the HCLV group received 0.4 mL/kg of 0.25% ropivacaine. Both groups received the same amount of ropivacaine (1 mg/kg). The primary outcome measure was the face, leg, activity, cry, consolability (FLACC) scale score at post-anesthetic care unit (PACU; T1). FLACC scale score at T1 was significantly lower in the HCLV group (2.91 versus 1.43; mean difference, −1.49; 95% confidence interval, −0.0245 to −2.936; *p* = 0.0464). FLACC scale scores one hour and six hours after the surgery were not different between the two groups. This study reports better post-operative analgesic effects after unilateral open inguinal hernia repair with 1 mg/kg of 0.25% ropivacaine than 1 mg/kg of 0.15% ropivacaine at PACU.

## 1. Introduction

Inguinal hernia is a common congenital disease among the pediatric population, and its management requires a surgical repair. To alleviate post-operative pain and discomfort, multimodal analgesic techniques, including the administration of opioids, non-opioid analgesics, neuraxial nerve blocks, and peripheral nerve blocks are widely used [1]. With ultrasound becoming the standard of care for daily practice of anesthesia and analgesia, peripheral nerve blocks are performed more frequently for post-operative analgesia. Among the various peripheral nerve blocks, the transverse abdominis plane (TAP) block can provide safe and optimal analgesia, especially after upper and lower abdominal surgical procedures [2,3,4]. Moreover, TAP block can be safely performed, even in neonates [5]. In infants, injected local anesthetics have a larger non-protein-bound fraction and greater volume of distribution, implying that the amount of local anesthetics required for a single dose nerve block is similar to that required in adults [6]. However, the optimal dose and concentration of local anesthetics, required for TAP block in the pediatric population, are not well defined, and potentially toxic doses are frequently administered to children [7]. For a given amount of anesthetic, the analgesic properties may vary with different combinations of volume and concentration [8]. Identifying the right combination of concentration and volume of anaesthetic can enable the administration of safe and optimal amount of local anesthetic for inducing regional analgesia in children. However, limited studies have compared the analgesic efficacy with different combinations of volume and concentration of anesthetics.

The primary goal of the current study was to compare the post-operative analgesic efficacy of TAP block between low-concentration high-volume (LCHV) and high concentration low volume (HCLV) combinations of local anesthetic. We hypothesized that, children in the LCHV group (1 mg/kg of 0.15% ropivacaine (0.67 mL/kg)), would experience less post-operative pain than those in the HCLV group (1 mg/kg of 0.25% ropivacaine (0.4 mL/kg)).

## 2. Experimental Section

This study was a prospective, randomized, double-blinded clinical trial. The study protocol was approved by the institutional review board of the Saint Mary’s Hospital, Seoul, Korea (KC16OASI0658), and was registered with the Clinical Research Information Service (http://cris.nih.go.kr; KCT0002176). Informed consent was obtained from the patients’ parents or legal guardians.

We collected data from a single tertiary hospital on the pediatric patients who underwent elective open unilateral inguinal herniorrhaphy, between November 2016 and October 2017. The inclusion criteria included patients aged between 0 years and 7 years, with an American Society of Anesthesiologists physical status of I. Patients with any other comorbidities, history of surgery, simultaneous multiple operations, and abnormalities on pre-operative laboratory evaluations were excluded.

During the study period, eighty-two pediatric patients underwent open herniorrhaphy for treating inguinal hernia. Five children were older than seven years. Seven children underwent another operation simultaneously. Ten children had bilateral inguinal hernia. The parents of fifteen children refused to provide consent for participation. One child was not a citizen of Korea, and was, therefore, excluded from the study because of the language barrier (Figure 1).

Forty-four patients were randomly assigned to the two groups (Figure 1), i.e., the low-concentration, high-volume (LCHV) group and the high-concentration, low-volume (HCLV) group. We used a web-based random assignment generator (https://www.graphpad.com/quickcalcs/randomize2/) to randomly allocate patients to the two groups. The results were maintained by an anesthesiologist who did not participate in the study.

During the course of the study, one patient from the HCLV group was excluded because of dosing errors, i.e., 0.7 mg/kg of 0.25% ropivacaine was injected for TAP block due to human error. Therefore, a total of forty-three patients successfully completed the study, and their data were analyzed.

All patients enrolled in the current study were anesthetized according to a standard protocol. All patients were sedated with 1.5 mg/kg of intravenous ketamine in the waiting room. After their transfer into the operating room, 6% sevoflurane, along with an intravenous bolus of 0.67 µg/kg of remifentanil, and 0.6 mg/kg of rocuronium was administered before endotracheal intubation. General anesthesia was maintained with one minimum alveolar concentration of sevoflurane and 0.05 µg/kg/min infusion of remifentanil. Ultrasound-guided TAP block was performed at the end of the surgery before awakening the patient from general anesthesia. We chose the posterior technique for TAP to provide prolonged analgesia [9].

Patients in the LCHV group received 0.67 mL/kg of 0.15% ropivacaine, while, those in the HCLV group received 0.4 mL/kg of 0.25% ropivacaine. Both groups received the same amount of ropivacaine (1 mg/kg). After complete emergence from anesthesia, the endotracheal tube was removed. Post-operative pain was assessed using the face, legs, activity, cry, consolability (FLACC) scale score at the following three time points: On arrival in the post-anesthesia care unit (PACU) (T1), one hour (T2), and six hours (T3) after surgery. T2 is approximately when the patient arrived at the general ward or day surgery center. T3 is when the effect of TAP block is about to diminish. The FLACC scale score is simple, well-validated, and is minimally affected by interpersonal variation [10].

The primary outcome measure of the intervention was at a FLACC scale score at PACU. The secondary outcome measure was the total fentanyl requirement at PACU, FLACC scale score at 1 h and 6 h after the surgery, and the presence of adverse events (nausea, vomiting, pruritis, seizure, arrhythmia, infection at the injection site, and bulging of the abdominal wall).

When the patient arrived at PACU, a PACU nurse continually measured the FLACC scale score every five minutes, until the FLACC scale score was less than 3. FLACC scale score >3 was considered as a failure of TAP block, and the PACU nurse administered 0.5 µg/kg of fentanyl initially, and subsequently, 0.25 µg/kg of fentanyl.

To avoid human bias, all TAP blocks were performed by a single anesthesiologist. Since the volume of injected drugs were different, it was possible that the practicing anesthesiologist could distinguish the groups. To ensure proper blinding, the drug was prepared by another anesthesiologist who did not participate in the study, and the syringe was covered with a non-transparent silicon plaster. Moreover, FLACC scale score was measured by PACU nurses who were blinded to the study protocol, and had no knowledge of the patients’ enrolment status. All the nurses were familiar with the FLACC scale, since it is a routine pain assessment tool at PACU in our institution.

GraphPad Prism version 7.03 for Windows (GraphPad Software, La Jolla, CA, USA, www.graphpad.com) was used for statistical analyses. A p-value less than 0.05 was considered to be statistically significant.

The results are presented as the mean ± standard deviation or as the median (interquartile range), as appropriate. Differences between the groups were analyzed using student’s t-test, Mann-Whitney U test, and Fisher’s exact test, when appropriate. The D ‘Agostino-Pearson normality test was used to test for the normal distribution.

## 3. Results

The patients’ demographic data are shown in Table 1. Data are presented as median (interquartile range) or frequencies. The youngest patient in the LCHV group was forty-two days old, and sixty-eight days old in the HCLV group. The oldest patients in both groups were seven years old. Admission to the hospital was mainly because of requests from parents.

The FLACC scale scores are shown in Table 2. FLACC scale score at T1 was significantly lower in the HCLV group (2.91 vs. 1.43; mean difference, −1.49; 95% confidence interval (CI), −0.0245 to −2.936; *p* = 0.0464). At T1, the failure rate of TAP block was 40.9% (9 out of 22 patients) in the LCHV group and 14.3% (3 out of 21 patients) in the HCLV group (Figure 2). However, the failure rate was not statistically different between the two groups (*p* = 0.0883; relative risk, 2.87; 95% CI, 0.99 to 8.95). FLACC score at T2 (0.73 versus 0.81; mean difference, 0.08; 95% CI, −0.46 to 0.62; *p* = 0.7603) and T3 (0.10 vs. 0.24; mean difference, 0.15; 95% CI, −0.26 to 0.55; *p* = 0.4661) were not significantly different between the two groups. At T2 and T3, patients in both groups showed a negligible pain sensation. All FLACC scale scores at T2 and T3 were less than three, except one. At T3, one patient of HCLV group cried when the nurse measured the FLACC scale score and the patient had a FLACC scale score of four. The mother of the patient thought her child was hungry and refused to take their medicine.

Fentanyl usage at PACU was not different between the two groups. (0.22 µg/kg vs. 0.07 µg/kg; mean difference, 0.14 µg/kg; CI, −0.0033 to 0.2877; *p* = 0.0551). (Figure 3). Three patients (two in the LCHV group; one in the HCLV group) showed FLACC scale scores of more than three at T1, and required repeat measurements of FLACC scale scores with additional fentanyl during PACU stay.

We also assessed the adverse events, associated with anesthetic administration, during the study. In the LCHV group, cases of vomiting and skin rash at the injection site were noted. The vomiting was controlled using a serotonin antagonist. The skin rash was accompanied by redness without any itching sensation, and resolved after a day. In the HCLV group, one case of sinus arrhythmia was noted on admission at PACU. The arrhythmia lasted five minutes before converting to sinus rhythm. The heart rate was between 120 and 140 beats per minutes, with stable blood pressures.

## 4. Discussion

The current study is the first to compare the analgesic effect of TAP block in pediatric open unilateral inguinal herniorrhaphy, with large-volume and low-concentration, and small-volume and high-concentration of ropivacaine anesthetic injection. In the current study, TAP block with 1 mg/kg of 0.25% ropivacaine successfully provided pain relief in 87% of the pediatric patients immediately after surgery, and reduced the need for fentanyl at PACU by 67%. Limited studies have investigated the effect of different concentrations and volumes of local anesthetics in regional anesthesia [11,12,13]. In children undergoing orchiopexy, caudal block, with a large volume of diluted ropivacaine, provided better quality and a longer duration of analgesia than a small volume of concentrated ropivacaine [8]. Similar results were reported with TAP block for an adult patient who underwent laparoscopic cholecystectomy. TAP block with a large volume and low concentration of bupivacaine was associated with smaller intra-operative requirement of remifentanil, less post-operative pain (at 20 min, 12 h, and 24 h after the procedure), and reduced requirement for post-operative analgesics [14]. However, in interscalene brachial plexus block for arthroscopic shoulder surgery in adults, a high concentration and lower volume of ropivacaine provided a faster onset of analgesia, although post-operative analgesic effects did not improve [15].

In the current study, there was a high rate of incomplete analgesia in the LCHV group. A possible explanation for this observation is the late onset of action of the low concentration of ropivacaine. The study on interscalene block with ropivacaine showed faster onset with the administration of 0.5% ropivacaine than with 0.25% ropivacaine [15]. We performed a posterior TAP block at the end of the surgery, and it is possible that 0.15% of ropivacaine was not fully effective at T1. The onset of posterior TAP block has not been studied properly [9]. On the contrary, the possibility remains that 0.15% ropivacaine (0.67 mL/kg) provided inadequate post-operative analgesia after unilateral, open herniorrhaphy in children.

This study has a few limitations. First, we did not fully implement a multi-modal method for analgesia. During the surgery, remifentanil was the sole analgesic agent, and its administration was discontinued at the end of the surgery. Since only intravenous fentanyl was administered after assessing the patient, remifentanil-induced hyperalgesia may have influenced the FLACC score at T1. However, the post-operative pain was well-controlled, since both the groups demonstrated complete pain relief at T2, and did not need additional analgesics during the remaining hospital stay. Second, because all patients received ketamine and sevoflurane, the occurrence of post-operative delirium or agitation may have affected the results. However, we did not assess the incidence of post-operative delirium. Last, although we tried our best to randomize the patients, the sample size of the present study was small, and therefore, the presence of bias cannot be ruled out. In particular, the failure rate of TAP block, and the need for fentanyl at PACU, could not be analyzed properly because of the small sample size.

Although, a higher dose of local anesthetics for regional nerve block provides prolonged relief from post-operative pain and decreases the need for rescue analgesics [12,13], it is worthwhile to investigate the optimal concentration and volume for the same dosage of local anesthetics owing to a perennial risk of overdose, due to local anesthetics, especially in pediatric patients.

## 5. Conclusions

The current study shows that the post-operative analgesic effects on the PACU after unilateral open inguinal hernia repair are better with 1 mg/kg of 0.25% ropivacaine, than with 1 mg/kg of 0.15% ropivacaine. Further, better-designed, large, randomized controlled trials are required to identify the optimal volume and concentration of local anesthetics for TAP block in children.

## Figures and Tables

**Figure 1 jcm-08-01133-f001:**
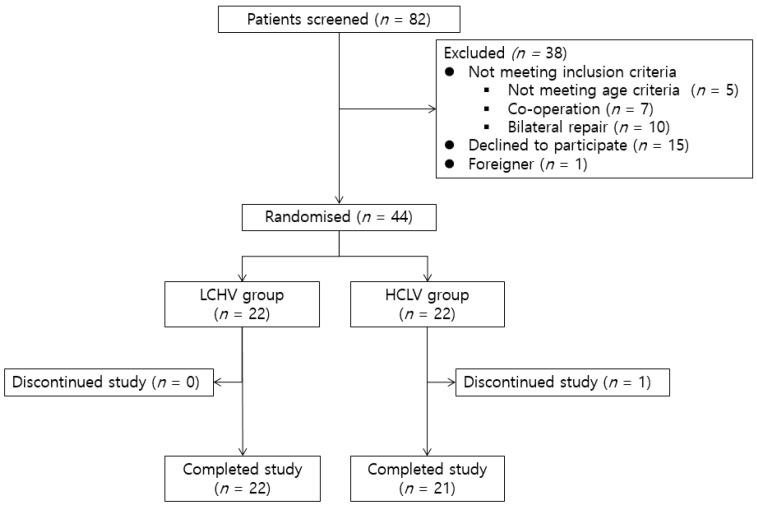
Consort Diagram of the study. LCHV, Low-concentration high-volume; HCLV, high-concentration low-volume.

**Figure 2 jcm-08-01133-f002:**
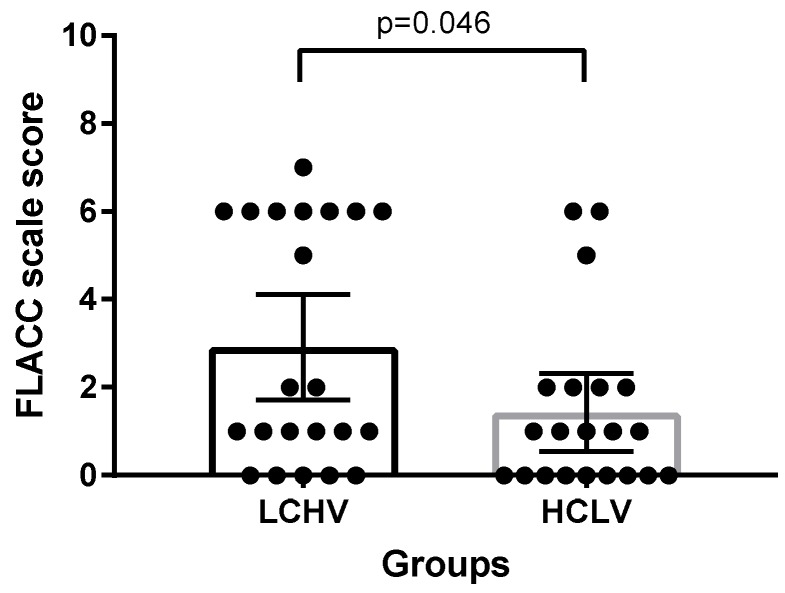
Face, leg, activity, cry, consolability scale score at post-anesthetic care unit. LCHV, Low-concentration high-volume; HCLV, high-concentration low-volume.

**Figure 3 jcm-08-01133-f003:**
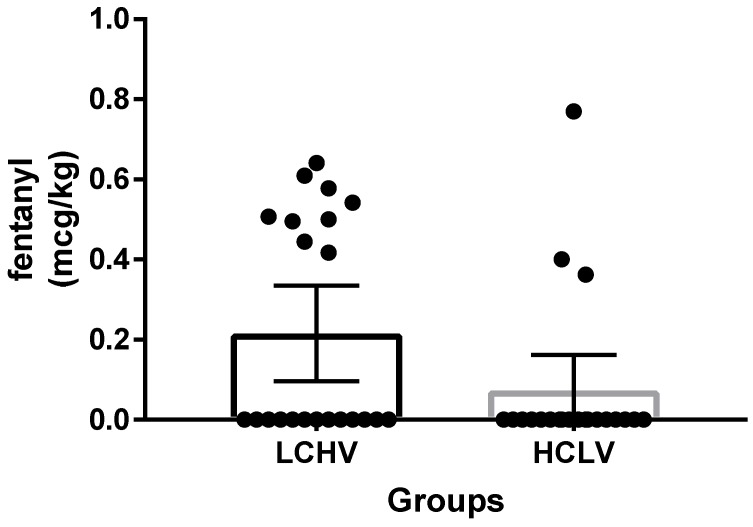
Fentanyl usage at post-anesthetic care unit. LCHV, Low-concentration high-volume; HCLV, high-concentration low-volume.

**Table 1 jcm-08-01133-t001:** Demographic data.

	LCHV	HCLV
Age (year)	3.5 (0.98–4.25)	2.25 (1.29–4.5)
Gender (male/female)	16/6	10/11
Ambulatory/Admission	18/4	14/7
Height (cm)	104.1 (74.7–110.3)	92 (77.2–102.0)
Weight (kg)	16.5 (9.1–19.5)	13.0 (10.3–15.3)
Operation Side (Left/Right)	10/12	10/11

LCHV, Low-concentration high-volume; HCLV, high-concentration low-volume.

**Table 2 jcm-08-01133-t002:** FLACC scale score.

	LCHV	HCLV	*p*-Value
T1	2.91 ± 2.706	1.43 ± 1.938	0.0464 (*)
T2	0.73 ± 0.827	0.81 ± 0.928	0.7603
T3	0.09 ± 0.294	0.24 ± 0.889	0.4661

LCHV, Low-concentration high-volume; HCLV, high-concentration low-volume; T1, patient arrived at PACU; T2, one hour after surgery; T3, six hours after surgery; (*), *p* < 0.05.

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
