# Peer review of "Comparison between Post-Operative Analgesic Efficacy of Low-Concentration High-Volume and High-Concentration Low-Volume Combinations of Ropivacaine for Transverse Abdominis Plane Block in Pediatric Open Inguinal Hernia Repair"

_jcm, 2019, doi:10.3390/jcm8081133_

Round 1

Reviewer 1 Report

The paper describes a fase 2 study, comparing different concentrations of ropivacaine in children undergoing hernia surgery under TAP block. The study is adequately conducted and the manuscript well-written.

Comments:

1.       My main concern is the way the results are presented. I would greatly appreciate a separate table, where all outcome measures are presented. It is difficult to understand the course of the FLACC score. In the methods section, it is stated that the FLACC score was measured every five minutes, but in the results section only three measures are mentioned. What does T1, T2 and T3 mean? The way it is presented, I also get the impression that the score was lower at T1 in the HCLV group (2.91 vs 1.43), but that the relationship was reversed at T2 (0.73 vs 0.81) and T3 (0.10 vs 0.24).

2.       The study is based on repeated measurements, without stating which of the measurements was the primary outcome. Did they consider a Bonferroni correction or any other method for adjusting for multiple comparisons?

3.       The acknowledgement section seems to include a text that was meant to be replaced before submission.

Author Response

We show our gratitude for thorough review of the article.

We revise the manuscript as you recommended.

Reviewer 2 Report

In their manuscript, entitled Comparison between the post-operative analgesic efficacy of low-concentration high-volume and high-concentration low-volume combinations of ropivacaine for transverse abdominis plane block in pediatric open inguinal hernia repair, the authors are presenting the results of a randomized clinical trial comparing two different local anesthetic volume/concentration regimens for TAP blocks in infants.

The overall idea of the study is interesting. The description of the study protocol and the CONSORT chart are good. The description of the results is adequate; the discussion is well-written and puts the current results into context. The limitations of the study are admitted and discussed properly.

I only have some very minor issues, which I would like the authors to address before I can fully endorse the publication of this manuscript:

Minor:

Abstract: Please use “ml” instead of “cc”

Sometimes “kilogram” is abbreviated as “kg”, sometimes not. Please unify.

Introduction, L41: “adult” should be changed to “adults”.

Experimental section, L95: The sentence starting with “After fully …” seems a little strange. Maybe it could be changed to “After complete emergence from anesthesia, the endotracheal tube was removed.”

Description of statistics, LL 119-121: Based on the description of the statistical methods used, I assume that an assessment of normal distribution has been undertaken prior to the actual testing. Please add the information, which method has been used to test for normal Distribution.

Author Response

(The authors gave the same response as above.)
